SCIENTIFIC CORRESPONDENCE

 

# Comment on 'Valid molecular dynamics simulations of human hemoglobin require a surprisingly large box size'

**Vytautas Gapsys, Bert L de Groot***

Computational Biomolecular Dynamics Group, Max-Planck Institute for Biophysical Chemistry, Göttingen, Germany

**Abstract** A recent molecular dynamics investigation into the stability of hemoglobin concluded that the unliganded protein is only stable in the T state when a solvent box is used in the simulations that is ten times larger than what is usually employed (El Hage et al., 2018). Here, we express three main concerns about that study. In addition, we find that with an order of magnitude more statistics, the reported box size dependence is not reproducible. Overall, no significant effects on the kinetics or thermodynamics of conformational transitions were observed.
DOI: https://doi.org/10.7554/eLife.44718.001

## Introduction

A surprising effect of the simulation box size was recently reported for spontaneous hemoglobin transitions from the T to the R state in molecular dynamics simulations of deoxy hemoglobin (*El Hage et al., 2018*). It was reported that the T state was stable for a period of 1 microsecond only in a relatively large simulation box with a cube edge length of 15 nm, whereas in smaller simulation boxes spontaneous transitions to the R state were observed on this timescale. Based on this, thermodynamic stabilization of the T state in the larger box was concluded, attributed to a change in the hydrophobic effect due to a change in the box size.

Here, we express three main concerns about the presented analyses. In addition, we present a systematic study of box size dependence in a variety of systems including hemoglobin, in which we observe no significant effect of the simulation box size on the kinetics or thermodynamics of conformational transitions.

## Results

### Statistics

First, in the work by El Hage et al. for each of the four studied box sizes, a single MD simulation was carried out and a transition (or lack thereof) was reported. Based on the single data points, an estimate of the uncertainty in the reported transition times is lacking, impairing an assessment of the statistical significance of the observed differences. The statistical significance of the concluded box size dependence has thus not been shown. In redoing the simulations presented by *El Hage et al. (2018)* using multiple replicas (20 for the 9 and 15 nm setup, 10 for the 12 nm setup, we find that there is considerable scatter in the transition times obtained in single MD simulations (see *Figure 1*). For example, we find that in the 9 nm box a substantial fraction does not undergo the transition within 1 μs, rendering the absence of a transition in the single simulation of the 15 nm box in the *El Hage et al. (2018)* study statistically insignificantly different from those observed for the 9 nm box. This is further underscored by the fact that out of 20 simulations we set up for the 15 nm system, 11 made the transition within 400 ns and 14 within 1 μs (*Figure 1C*). Consistent with this result,

***For correspondence:**
bgroot@gwdg.de

**Competing interests:** The authors declare that no competing interests exist.

we find that the distributions of RMSD values to the T and R state crystal structures monitored at 400 ns and 1 µs of simulation time are highly similar for the studied box sizes (*Figure 1A,B*). Based on the statistics from our multiple simulations, we can quantify the probability for a box size-effect on the transition kinetics of hemoglobin. Specifically, we have estimated the probability of a slow-down in the 15 nm box, as claimed by *El Hage et al. (2018)*. For a relatively moderate effect of a factor of 2 slower kinetics, this amounts to a probability of 0.0026, for a factor of 10 it amounts to 4.7e-6. We can thus reject the box size dependence hypothesis with a relatively large statistical margin.

As an independent validation of this result, we have monitored possible back-transitions to the T state after a T to R transition using the El Hage et al. simulation setup. We have not observed a full back-transition in any of the simulated systems. We only observe occasional partial transitions back to the T state (see also *Figure 1—figure supplement 1*), independent of the box size. This is consistent with the finding above that there is no detectable stabilization of the T state over the R state in the 15 nm box.

## Hydrophobic effect

Second, the reported box size dependence was attributed to the hydrophobic effect, and analysed using solvent-solvent hydrogen bonds, solvent radial distribution function and water diffusion constants (*El Hage et al., 2018*). Here, we are concerned that the presented analyses report not only on a possible hydrophobic or hydrodynamic effect, but also on a dilution effect caused by the different ratio of protein to water in differently sized boxes. Indeed, if we reanalyze the solvent radial

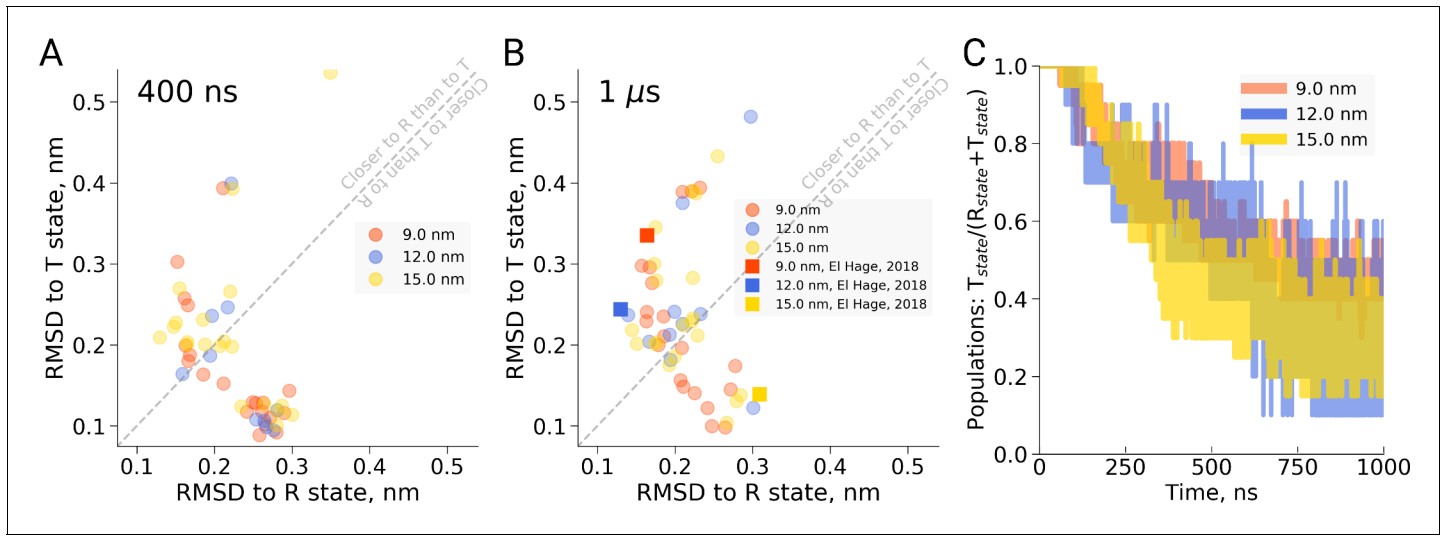

**Figure 1.** Hemoglobin simulation analysis. (**A**) RMSD of the simulation snapshots after 400 ns to the crystallographic T (2dn2) and R (2dn3) states. (**B**) RMSD of the simulation snapshots after 1 µs. Note that the simulations reported by *El Hage et al. (2018)* (squares) scatter indistinguishably from the other replicas. (**C**) Fraction of trajectories remaining in T for simulations in the boxes of different size. Simulations were counted to have left the T state if more than half of the distance from the T state had been covered, as projected onto the difference vector between the T and R crystallographic states. Each simulation reached 1 µs and comprised the following replicas: 20 for 9 nm and 15 nm boxes, 10 for 12 nm box.

DOI: https://doi.org/10.7554/eLife.44718.002

The following source data and figure supplements are available for figure 1:

**Source data 1.** Source files for RMSD and transition analysis.
DOI: https://doi.org/10.7554/eLife.44718.005

**Figure supplement 1.** RMSD values for the snapshots of hemoglobin simulations in 9 nm (left), 12 nm (middle) and 15 nm (right) boxes.
DOI: https://doi.org/10.7554/eLife.44718.003

**Figure supplement 1—source data 1.** This figure is made from the same data as *Figure 1*.
DOI: https://doi.org/10.7554/eLife.44718.004

distribution function (*Figure 2A*) or water-water hydrogen bonds (*Figure 2B*) and normalize based on the analysis volume (which we keep constant for the differently sized boxes) rather than by the box volume as was done originally, we find no significant box size effect. A similar dilution effect is observed for the diffusion constant. Averaging over the complete box represents a weighted average over restricted and bulk waters. For larger boxes, more water molecules are unrestricted by the protein and therefore behave bulk-like, yielding a larger weight of bulk water to the average. To disentangle this dilution averaging effect from possible box size effects, we estimated average diffusion constants for the 12 and 15 nm boxes from that of the 9 nm box, reweighting solely with the additional number of water molecules in the larger boxes and applying the bulk value of $5.9 \cdot 10^{-5} cm^2$/s from El Hage et al. As can be seen in *Figure 2C*, the resulting diffusion constants obtained this way

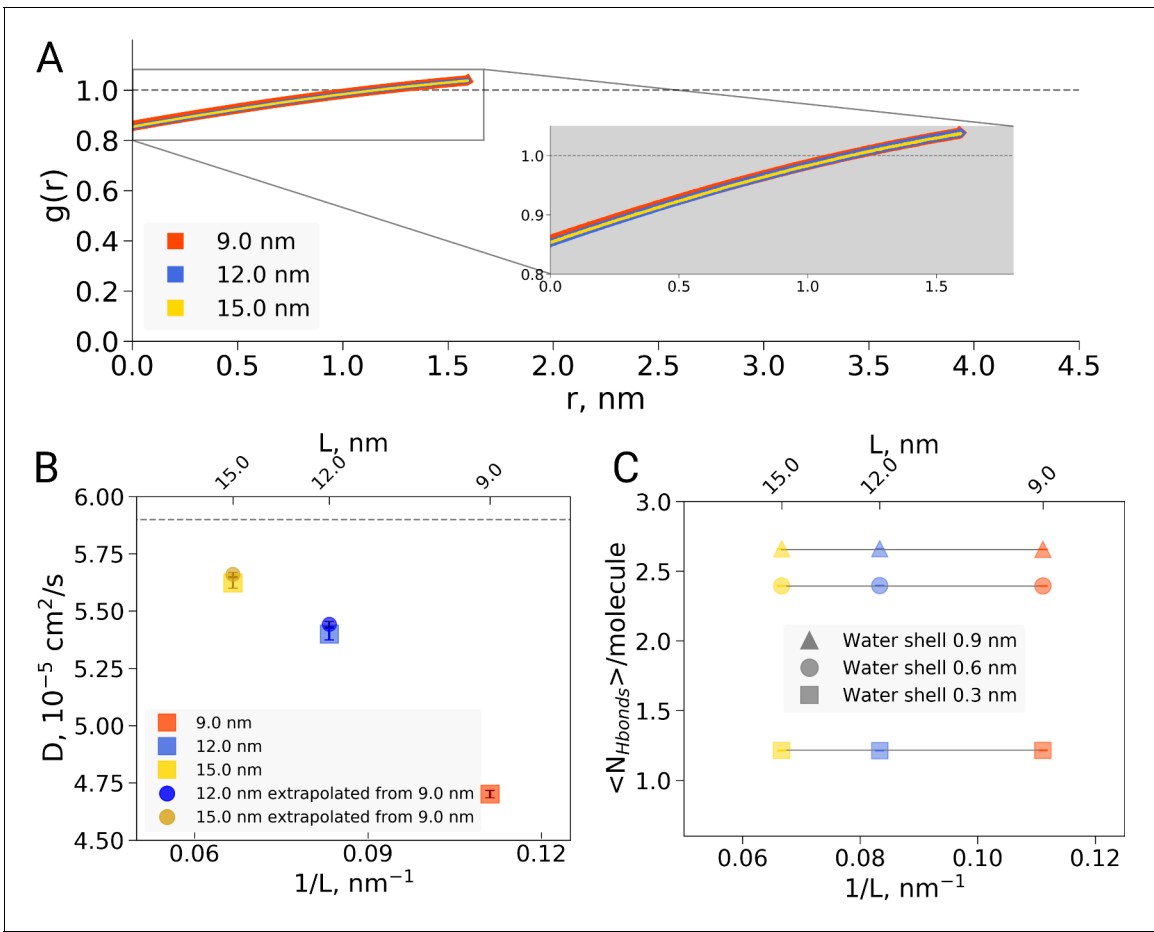

**Figure 2.** Radial distribution function (RDF), water diffusion and hydrogen bonds in hemoglobin simulations. (A) RDF for water oxygen atoms was calculated for the simulations in the boxes of different sizes. An equivalent volume for estimating the RDFs was used for every box size by cutting out a cubic box with an edge of 9 nm around the protein. (B) The water diffusion coefficient is plotted as a function of the box edge length. The square symbols mark the results obtained directly from simulations. The circles denote diffusion coefficients calculated from the value obtained from the 9 nm box simulations by means of extrapolation by adding a corresponding number of bulk water molecules (see main text). (C) Average number of water-water hydrogen bonds (cutoff value for the donor-acceptor distance of 0.33 nm) for three solvation shells around the protein: 0.3, 0.6 and 0.9 nm.
DOI: https://doi.org/10.7554/eLife.44718.006

The following source data and figure supplements are available for figure 2:

**Source data 1.** Source files for RDF, diffusion and hydrogen bond analysis.
DOI: https://doi.org/10.7554/eLife.44718.009

**Figure supplement 1.** Radial distribution function (RDF) and water diffusion for ubiquitin simulations with the Charmm36m force field.
DOI: https://doi.org/10.7554/eLife.44718.007

**Figure supplement 1—source data 1.** Source files for RDF, and diffusion analysis of ubiquitin.
DOI: https://doi.org/10.7554/eLife.44718.008

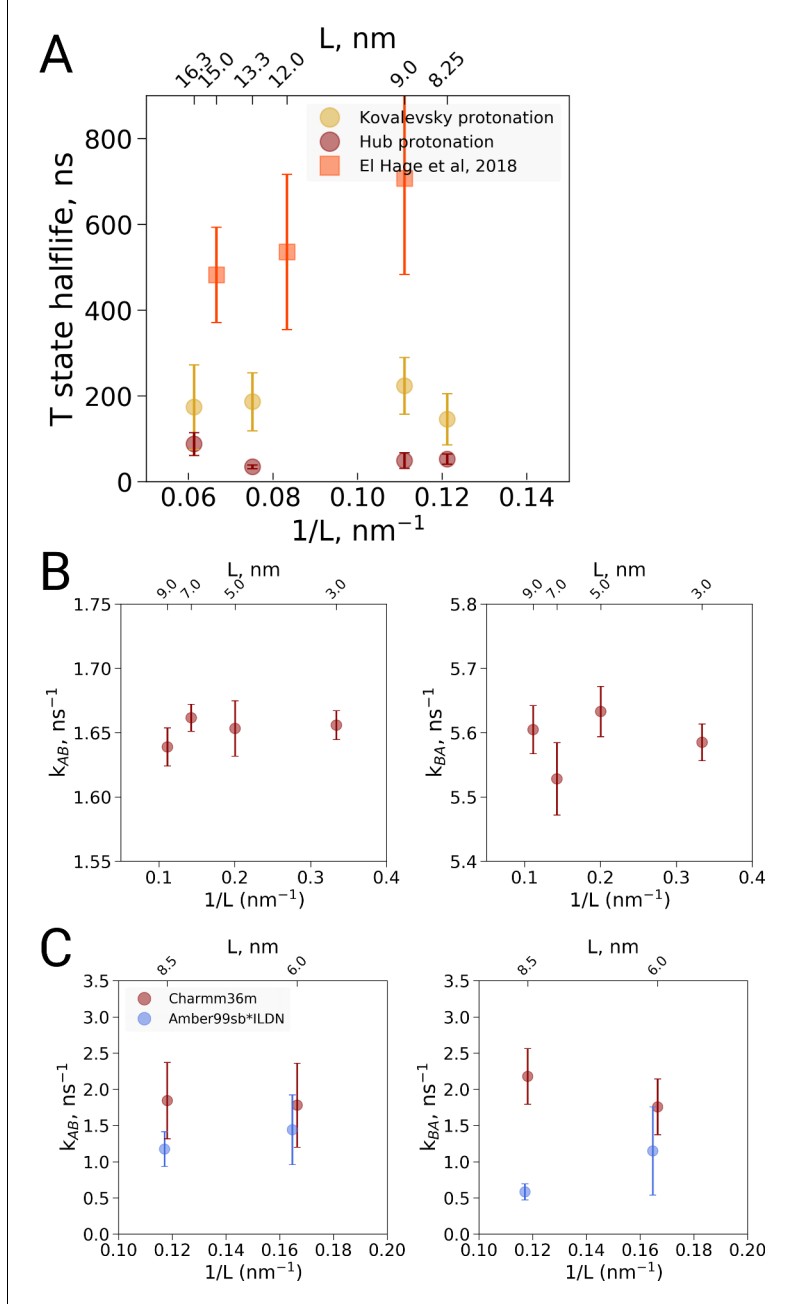

**Figure 3.** Kinetic analysis of hemoglobin, ubiquitin and alanine dipeptide. (**A**) Hemoglobin: average transition time from T to R, expressed as the time in which half of the replicas displayed the T to R transition, as a function of the simulation box size. Three different system setups were used for simulations. The Kovalevsky and Hub setups differ in the protonation of hemoglobin (***Vesper and de Groot, 2013***), both model the iron-proximal histidine interaction as a covalent bond. The El Hage setup is obtained from El Hage et al and thus identical to the one used in ***El Hage et al. (2018)***, where the iron-histidine were not covalently bound. Note that the difference in size of the error bars is explained by the difference in transition times: larger transition times are accompanied by larger absolute uncertainties. (**B**) Transition rates between the two minima in ubiquitin: rates from A to B (left) and from B to A (right). The reaction coordinate with the minima is depicted in ***Figure 4E,F***. (**C**) Transition rates between the two minima in alanine dipeptide: rates from A to B (left) and from B to A (right). The reaction coordinate with the minima is depicted in ***Figure 4B,C***.
DOI: https://doi.org/10.7554/eLife.44718.010

The following source data is available for figure 3:

**Source data 1.** Source files for kinetics analysis.
DOI: https://doi.org/10.7554/eLife.44718.011

are remarkably close to the ones originally computed for the 12 and 15 nm sized boxes, indicating that the observed differences in average diffusion constant can be largely explained by the dilution effect and thus do not reflect a box size effect. This is consistent with earlier results from *Yeh and Hummer, 2004* who showed that the box size dependence on water diffusion is small for box sizes larger than 4 nm. The dilution effect is evident in other systems as well: *Figure 2—figure supplement 1* demonstrates the RDF and diffusion coefficient dependence on the protein-to-water ratio in ubiquitin simulations. Note that ubiquitin is significantly smaller than hemoglobin and therefore the hydrophobic effect should be expected to be significantly smaller, following the argumentation of *El Hage et al. (2018)*. The fact that nevertheless we see a similar box size effect on the (unnormalized) RDF and diffusion constant confirms that the underlying cause is the dilution effect rather than a box size effect on the hydrophobic effect.

As the inherent, unavoidable dilution effect associated with increasing the box size is sufficient to explain the observed data, this implies there is no evidence to justify assuming an additional effect on protein solvation or the hydrophobic effect. This thus removes the justification for Figures 3, 4 and 5 of *El Hage et al. (2018)*. It is important to note that this does not preclude the hydrophobic effect hypothesis: however, the current data do not indicate that protein solvation or the hydrophobic effect changes significantly with the simulation box size for the studied range.

## Kinetics rather than thermodynamics

Third, we argue that the readout of transition times from T to R as applied is insufficient to allow for conclusions on the thermodynamic stability of the T state. According to rate theory, the transition times are primarily related to the activation barrier and the attempt frequency of the transition. Therefore, longer transition times could either be caused by a lowering of the attempt frequency or an increased barrier. In turn, an increased barrier (e.g. to leave the T state) can be achieved by a higher thermodynamic stability (lowering the well of the T state) but equally well by solely increasing the barrier and thus leaving the thermodynamic equilibrium between T and R unaltered. Without addressing the barrier and the attempt frequency, transition times to leave a state are thus insufficient to draw conclusions about the thermodynamic stability of that state. We thus argue that the presented transition times report primarily on the kinetics of the T to R transition rather than the thermodynamic difference between the T and R states.

## Kinetics as a function of box size

To investigate if the box size may affect transition kinetics for other systems, we have systematically investigated conformational transitions for a variety of additional systems including alternative setups of deoxy hemoglobin, each based on multiple independent trajectories.

We carried out MD simulations of deoxy hemoglobin in the Charmm36m force field for four different box sizes ranging from 8.25 nm to 16.3 nm for two different sets of protonation states for the histidine residues, starting from the T state (PDB entry 2dn2). In addition, as described above, simulations based on the El Hage et al. setup were carried out: ten replicas of each state were simulated for statistics, except for the El Hage et al. setup with box sizes of 9 and 15 nm for which we did 20 replicas. Spontaneous transitions towards the R state were observed under each of the simulated conditions (*Figure 3A*). Histidine protonation states as used by *Hub et al. (2010)* were found to lead to generally faster transitions than histidine protonation as suggested by *Kovalevsky et al., 2010*, in line with earlier simulations using the GROMOS96 43A1 force field (*Van Gunsteren et al., 1996*; *Vesper and de Groot, 2013*). Interestingly, the setup of El Hage et al. leads to generally slower transitions than either of these two protonation states. No dependence on the simulation box size was observed for the transition times for any of the tested setups (*Figure 3*).

### Factors influencing the T to R transition kinetics

In a preliminary investigation into the cause of the generally slower transitions in the El Hage et al. setup, we noted that the histidine protonation states were virtually identical to the *Hub et al. (2010)* setup and therefore are unlikely responsible for the slower transition times. The only difference in protonation state concerns His72, which is protonated on the delta position in both $\alpha$ subunits in the Hub setup, whereas it is protonated on the delta position on the $\alpha-1$ subunit and on the epsilon position on the $\alpha-2$ subunit in the El Hage et al. setup. Since $\alpha$-His72 is exposed to the solvent, its

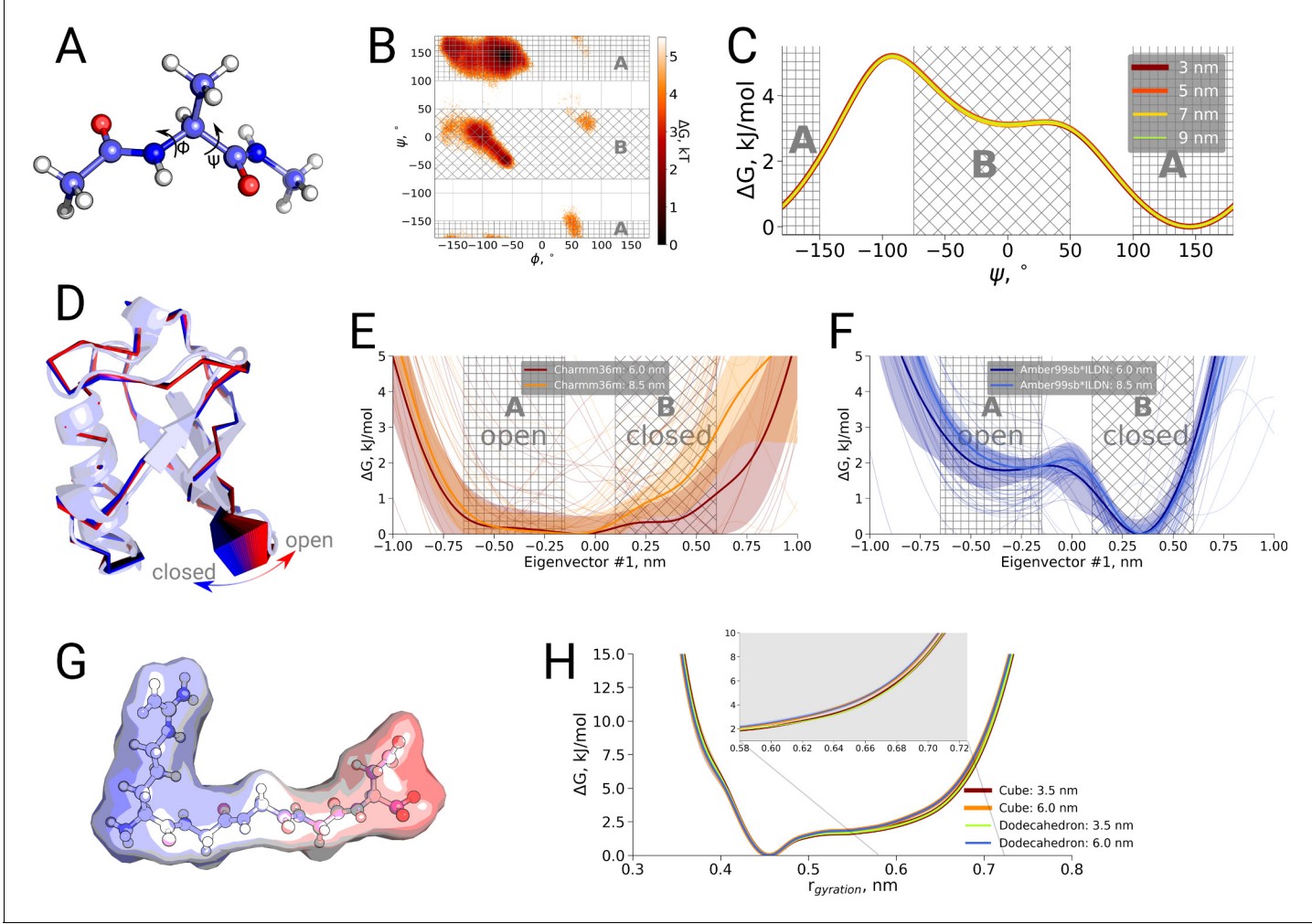

**Figure 4.** Thermodynamic analysis of hemoglobin, ubiquitin and RGGGD peptide. (**A**) Structure of the alanine dipeptide. (**B**) Free energy surface for alanine dipeptide simulated in the smallest box projected onto the backbone dihedrals marked in the panel (**A**). (**C**) Free energy profile along the $\psi$ dihedral angle. (**D**) Structure of ubiquitin and an interpolation of the motion along the principal component with the largest variance (pincer mode). (**E**) Free energy profile projected on the pincer mode for ensembles generated with the Charmm36m force field. (**F**) Free energy profile for the Amber99sb*ILDN force field. (**G**) Structure of the RGGGD pentapeptide with the surface color corresponding to the surface charge: blue - positive, red - negative. (**H**) Free energy profiles for the pentapeptide projections as a function of the radius of gyration.
DOI: https://doi.org/10.7554/eLife.44718.012

The following source data is available for figure 4:

**Source data 1.** Source files for thermodynamics analysis.
DOI: https://doi.org/10.7554/eLife.44718.013

protonation is unlikely to affect the T to R transition. Instead we identified two other differences that appear to slow down the transition. First, in the El Hage et al. setup an angular center of mass motion removal was employed on the protein, whereas in the Hub et al. and Vesper et al. a linear center of mass motion removal for the whole system was used. Second, whereas in the *Hub et al. (2010)* and *Vesper and de Groot, 2013* setups a covalent bond was modeled between the iron and the epsilon nitrogen of the proximal histidine for all four subunits, this interaction was modeled with non-bonded interactions in the *El Hage et al. (2018)* setup. Note that another difference in setup is the use of cubic and rhombic dodecehedral boxes. Based on the current data, the box geometry does not appear to have a major effect, but more systematic analyses would be required for a definitive answer.

### Kinetics of conformational transitions in other systems

To further examine possible kinetic effects caused by the box size we looked at two additional systems: alanine dipeptide and ubiquitin (*Figure 3B,C*). For alanine dipeptide, we counted transitions for the $\psi$ dihedral, for ubiquitin transitions along the principal 'pincer' mode (*Lange et al., 2008*) were investigated. Also for these systems, no systematic effect of the box size on the transition kinetics was observed.

### Thermodynamics as a function of box size

Finally, we also investigated a possible box size influence on the thermodynamics of conformational transitions for alanine dipeptide, ubiquitin and the pentapeptide RGGGD. These systems have sufficiently small transition barriers to derive converged free energy profiles from unbiased simulations on the microsecond timescale. We constructed the pentapeptide RGGGD with the termini in their zwitterionic form, and thus carrying two positive charges on the N-terminus and two negative charges on the C-terminus, as a particularly sensitive test to long-range electrostatic effects on the conformational dynamics. In this case, no ions were added to the box, to avoid any screening effects. For the three systems, only minor effects, if any, of the box size on the obtained free energy profiles were observed (*Figure 4*).

## Conclusions

In conclusion, we have shown that the observation of *El Hage et al. (2018)* that the stability of hemoglobin in the T state in the largest box of 15 nm was not significantly different from the other investigated box sizes. In addition, a re-analysis of solvent hydrogen bonds, solvent radial distribution function and diffusion constants as a possible underlying hydrophobic or hydrodynamic effect showed that, when normalized to the analysis volume rather than the box volume, no significant box size effects were observed. Moreover, an analysis of the conformational transition kinetics in three hemoglobin setups as well as two additional systems showed no significant box size effects. Similarly, for three systems that allowed the investigation of thermodynamic effects, no significant box size effects were observed.

The results from hemoglobin simulations indicate that under the simulated conditions the R state is favored over the T state, with the histidine protonation states affecting the transition statistics, consistent with the Bohr effect (*Zheng et al., 2013*) and previous simulations (*Hub et al., 2010*; *Vesper and de Groot, 2013*).

Our analyses show that the dependence of the thermodynamic and kinetic properties on the box size is well within the level of statistical fluctuations for the observed statistics of 10–20 independent simulations at the microsecond timescale. The differences arising due to the different force fields, starting structures and chaotic nature of the simulations in this regime were found to be larger than the box size induced effects for the investigated systems.

## Materials and methods

### Simulation setup

Three variants of the hemoglobin simulation setup were prepared: 1) El Hage (*El Hage et al., 2018*), 2) Hub (*Hub et al., 2010*) and 3) Kovalevsky (*Kovalevsky et al., 2010*). The setups differed in the protonation state assignment and the protein-heme interaction treatment (see main text for details). The molecular dynamics simulations for the El Hage setup were performed in cubic boxes and used the parameter set and initial structure obtained from El Hage et al. for the 9 nm box. The setup is therefore identical to that reported in *El Hage et al. (2018)*. The 12 and 15 nm setups were generated by adding an additional shell of water and ions around the 9 nm box. These generated 12 and 15 nm boxes behave indistinguishably from those reported by El Hage et al. (*Figure 1*). For the Hub and Kovalevsky setups dodecahedral simulation boxes were used in combination with the molecular dynamics parameters compatible with the Charmm force field. The Charmm36m (*Huang et al., 2016*) force-field was used for all the simulations. Hemoglobin was solvated with TIP3P water, Na$^+$ and Cl$^-$ ions were added to neutralize the system and reach 150 mM salt concentration. Equations of motion were integrated with a 2 fs time step. The temperature was kept at 300 K by means of the

velocity rescaling thermostat (*Bussi et al., 2007*) with a time constant of 1.0 ps. All atoms in the system were assigned to one temperature coupling group. The Parrinello-Rahman barostat (*Parrinello and Rahman, 1981*) was used to keep the pressure at 1 bar (time constant 5.0 ps). Long range electrostatics were treated by means of PME (*Darden et al., 1993*; *Essmann et al., 1995*) with a real space cutoff of 1.2 nm, a fourier spacing of 0.12 nm and interpolation order of 4. Van der Waals interactions were cut off at 1.2 nm by smoothly switching off the force starting from 1.0 nm. Several noteworthy differences between the El Hage and other setup variants include a dispersion correction to energy and pressure which was used by El Hage et al, but was not employed for the other setups in our work. While El Hage setup used LINCS (*Hess et al., 1997*) to constrain all bonds, Hub and Kovalevsky setups constrained hydrogen atoms involving bonds only. The El Hage setup also removed angular center of mass motion of the protein during the simulation; this option was not used in other setups.

Simulations using the El Hage setup were performed in three cubic boxes with a cube edge of 9.0, 12.0 and 15.0 nm. For the size of 9.0 nm and 15.0 nm, 20 independent simulations of 1 µs were performed, while for the size of 12.0 nm, we used 10 simulations of 1 µs, in line with the rule of thumb by *Knapp et al. (2018)*. The Hub and Kovalevsky setup runs were performed in dodecahedral boxes which have volume equivalent to the cubic boxes with an edge of 8.25, 9.0, 13.3 and 16.0 nm. For each box 10 simulations were performed reaching in total 2.8, 2.0, 0.9 and 1.2 µs for every box, respectively, in the case of Hub setup and 5.0, 4.0, 2.3 and 2.6 µs for every box in case of Kovalevsky setup.

The alanine dipeptide, ubiquitin and RGGGD peptide simulations used the same simulation parameters as Hub and Kovalevsky hemoglobin setups. Alanine dipeptide was simulated in four cubic boxes with edges of 3.0, 5.0, 7.0 and 9.0 nm. For every box size 10 simulations 1 µs each were performed. Ubiquitin was simulated in two cubic boxes with edges of 6.1 and 8.5 nm. For ubiquitin 20 simulations 1 µs each were performed per box size: 10 runs starting from the 1ubq and 10 initialized with 1nbf structure. Furthermore, 20 additional ubiquitin simulations of 1 µs each were performed using Amber99sb*ILDN force field. RGGGD peptide was simulated in two cubic boxes with an edge length of 3.5 and 6.0 nm. For both box sizes 10 simulations of 1 µs were performed. In addition, the same simulation procedure for the RGGGD peptide was performed using dodecahedral boxes with the volume equivalent to the cubic boxes with an edge of 3.5 and 6.0 nm.

## Analysis

Root mean squared deviation (RMSD), radial distribution function (RDF), diffusion coefficient calculation and hydrogen bond counting were performed with the standard tools in the Gromacs (*Abraham et al., 2015*) library. Prior to calculating RDF for the boxes with an edge of 12.0 and 15.0 nm, a cubic box with an edge of 9.0 nm centered at the protein was cut out from the original boxes. This ensured consistent normalization for all the box sizes. As for the hydrogen bond counting, the consistent normalization was ensured by considering only the waters around the protein in the shells at a distance of 0.3, 0.6 and 0.9 nm. For the box size dependence of the average diffusion constant, a bulk extrapolation based on the 9.0 nm box was used for the 12.0 and 15.0 nm boxes as described in the results section.

The error bars for hemoglobin halflife times, as well as alanine dipeptide and ubiquitin rate constants were calculated by means of bootstrap.

---

## Additional information

### Funding

| Funder | Grant reference number | Author |
| --- | --- | --- |
| European Commission | t H2020-EINFRA-2015-1-675728 | Bert L de Groot Vytautas Gapsys |

The funders had no role in study design, data collection and interpretation, or the decision to submit the work for publication.

---

### Author contributions
Vytautas Gapsys, Conceptualization, Software, Formal analysis, Investigation, Visualization, Methodology, Writing—original draft, Writing—review and editing; Bert L de Groot, Conceptualization, Software, Formal analysis, Funding acquisition, Investigation, Methodology, Writing—original draft, Writing—review and editing

### Author ORCIDs
Vytautas Gapsys (iD) http://orcid.org/0000-0002-6761-7780
Bert L de Groot (iD) https://orcid.org/0000-0003-3570-3534

### Decision letter and Author response
Decision letter https://doi.org/10.7554/eLife.44718.016
Author response https://doi.org/10.7554/eLife.44718.017

---

## Additional files

### Data availability
We have provided the raw data for all of the graphs in all of the figures as Source data. For each figure we are providing ReadMe files that include descriptions for generating the figures.

---

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
