## [Decision Letter]

Thank you for submitting your article "Comment on 'Valid molecular dynamics simulations of human hemoglobin require a surprisingly large box size'" for consideration by eLife. Your article has been reviewed by 2 peer reviewers, and the evaluation has been overseen by a Reviewing Editor (Yibing Shan), a Senior Editor (John Kuriyan), and the eLife Features Editor (Peter Rodgers). The following individual involved in review of your submission has agreed to reveal his identity: Wei Yang (Reviewer #1).

The reviewers have discussed the reviews with one another and the Reviewing Editor has drafted this decision to help you prepare a revised submission. Please aim to submit the revised version within two months.

Summary:

This comment by Gapsys and de Groot concerns the eLife paper 'Valid molecular dynamics simulations of human hemoglobin require a surprisingly large box size' by El Hage et al.

The comment reports that (a) its authors could not observe an effect of large box size on the T-R transition of hemoglobin in their own simulations, and it argues that (b) the analysis of El Hage et al. may be flawed and the authors therefore challenge the hydrophobic interpretation of the perceived box size effect. The reviewers think that the community of scientists using molecular dynamics simulations will appreciate the efforts by Gapsys and de Groot for their careful re-examination of the box size effect.

Essential revisions:

1) As a whole, the MD community understands relatively little about water box size effects on protein conformational transitions, especially on conformational transitions of large systems. The community's analyses on this topic so far have been mainly based on smaller systems, such as alanine dipeptide, ubiquitin, and the RGGGD pentapeptide. The possibility that the box size effect becomes pronounced only for large protein systems cannot be excluded, although it is far from established. The work by El Hage certainly fell short of performing enough (in the order of hundreds or more) repeats of each simulation to establish statistical significance for their observation. (This is acknowledged by El Hage et al in their response to the comment.) This comment reported substantially more repeats of the simulations (~10) than El Hage et al., but it also falls short of establishing the statistical significance of an opposite observation. The reviewers believe it is important for both parties to not overstate the case, and to clearly acknowledge that the box size effect remains a hypothesis, neither established nor convincingly precluded. Unless the authors think this is not the case, the comment should be revised to reflect this basic reality that neither parties in this debate knows the ultimate answer here.

2) The discussion on the distinction between thermal and kinetic stability in the first paragraph of the results section is not needed and should be removed. (The corresponding section in the reply by El Hage et al. should also be removed.) El Hage highlighted that the observed discrepancy between the experimental timescale of the T-R transition (seconds) and the much shorter simulation timescale of the transition. Although an underlying kinetic effect is also possible, it is not unreasonable to suggest that the thermodynamics of the simulation system is distorted. There is no indication that El Hage et al. are not aware of the distinction between these two scenarios. At this juncture it is a secondary issue to be concerned about whether the box size effect is of thermodynamic or kinetic nature, when the effect itself is under debate.

3) It is known, though rarely formally acknowledged, that when the simulation length is far shorter than the actual timescale of a molecular event, the simulation results are sensitive to the exact simulation setup. We note that the setups by El Hage et al and by Gapsys and de Groot are not identical, and the differences in the setups potentially involve several parameters (e.g., how the initial water box was constructed, how long the water molecules were equilibrated before the production runs, whether the protein system was restrained during the water equilibriation, and what types of baro-stat and thermo-stat were used et al.) To clarify these issues, please be explicit about the differences between your setups and the setups used by El Hage et al.

Also, please revise your manuscript to address the following comments in the response from El Hage et al.a) The comments in the sentence that starts "A possible problem with the comparisons in their figure 3…"b) The comments in the sentence that starts "We also note a difference in the magnitude of the error bars…"c) The comments in the paragraph that starts "When we were still collaborating…"

4) The comment showed that using a different analysis of the water behavior leads to a different conclusion. In the reply, El Hage et al. clarified that their attribution of the box size effect to the hydrophobic effect remains a hypothesis and it requires further investigation. Indeed, this hypothesis needs further examination by careful analyses of water structures and by extensive free energy calculations. The reviewers suggest that in the revision the authors make it clear that the hydrophobic effect hypothesis is not precluded, although the original analysis by El Hage can be improved.

5) The box size effect, if true, is necessarily a subtle one. The authors' simulations of small proteins without observing a box size effect do not serve as proof to preclude the box size effect, especially for large systems. The authors should make this important caveat clear in their discussions.

---

## [Author Response]

We repeat the reviewers’ points here in italic, and include our replies point by point, as well as a description of the changes made, in Roman.

Essential revisions:1) As a whole, the MD community understands relatively little about water box size effects on protein conformational transitions, especially on conformational transitions of large systems. The community's analyses on this topic so far have been mainly based on smaller systems, such as alanine dipeptide, ubiquitin, and the RGGGD pentapeptide. The possibility that the box size effect becomes pronounced only for large protein systems cannot be excluded, although it is far from established. The work by El Hage certainly fell short of performing enough (in the order of hundreds or more) repeats of each simulation to establish statistical significance for their observation. (This is acknowledged by El Hage et al in their response to the comment.) This comment reported substantially more repeats of the simulations (~10) than El Hage et al., but it also falls short of establishing the statistical significance of an opposite observation. The reviewers believe it is important for both parties to not overstate the case, and to clearly acknowledge that the box size effect remains a hypothesis, neither established nor convincingly precluded. Unless the authors think this is not the case, the comment should be revised to reflect this basic reality that neither parties in this debate knows the ultimate answer here.

REPLY: Please note that there is a logical difference between 'there is not sufficient evidence to conclude a box size effect' and 'there is no box size effect'. The two require entirely different statistical tests. In our comment we refer to the former, the reviewers suggest we do not provide the required statistics to conclude the latter. Please note that nowhere in our comment have we stated that there is no box size effect. We have merely stated that based on the data presented in the El Hage et al paper as well as in our own data we find no support for the box size hypothesis (we have not claimed 'an opposite observation'). The original suggestion of a box size effect thus requires the same statistical test as our assertion that there is not sufficient evidence to support that hypothesis. Against that background therefore it should be noted that our N=10-20 repeats carry substantially more statistical weight than the single N=1 trajectories reported by El Hage et al.

In the meantime we have collected additional statistics which in addition to showing that the El Hage et al. results were not significant, actually provide a sufficient basis for a quantitative estimate of the likelihood that the box size hypothesis is valid. Specifically, we estimated the probability that the 150A box shows a substantially longer transition time than the smaller boxes which the authors use to support the El Hage et al. claim "Valid Molecular Dynamics Simulations of Human Hemoglobin Require a Surprisingly Large Box Size". For a relatively moderate effect of a factor of 2 slower kinetics, this amounts to a probability of 0.0026, for a factor of 10 it amounts to 4.7e-6. We can thus, in fact, reject the box size dependence hypothesis with a relatively high statistical confidence (note that this does not take “hundreds or more” repeats. As suggested recently, also in this case on the order of 10 repeats appears suitable: DOI:10.1021/acs.jctc.8b00391). We have now incorporated this quantification in the updated manuscript. Instead of requesting us to "clearly acknowledge that the box size effect remains a hypothesis, not convincingly precluded", El Hage et al. should thus be requested to clearly state that there is insufficient statistical basis to support the box size hypothesis.

Finally, it is written in the decision letter “The work by El Hage certainly fell short of performing enough repeats of each simulation to establish statistical significance for their observation. This is acknowledged by El Hage et al in their response to the comment.” However, this is not the case. Specifically, the abstract of the El Hage et al. response that was sent to us states “the studies do not invalidate the conclusion that there is a significant box size effect” and the conclusion states “our simulations provide evidence that there is a box size dependence in hemoglobin simulations”. As we demonstrate in our comment, the results of El Hage et al. are not statistically significant (or, in the words of the editors/reviewers: “The work by El Hage certainly fell short of performing enough repeats of each simulation to establish statistical significance for their observation”). Therefore the results do not support the drawn conclusion of a box size dependence. This conclusion should thus be withdrawn from the response of El Hage et al, and rather should the lack of a statistical basis for the original paper (El Hage et al., 2018) be acknowledged.

Changes made to the manuscript:

Figure 1, panels B and C updated with the full 1 microsecond trajectories.

Figure 2, panels A, B and C updated with the full 1 microsecond trajectories

Figure 3, panel A updated with the full 1 microsecond trajectories.

Manuscript text: the text has been updated to account for the fact that now all our hemoglobin simulations have reached a length of 1 microsecond. In addition, we have added the quantification: “Based on the statistics from our multiple simulations, we can quantify the probability for a box size-effect on the transition kinetics of hemoglobin. Specifically, we have estimated the probability of a slowdown in the 15 nm box, as claimed by El Hage et al. For a relatively moderate effect of a factor of 2 slower kinetics, this amounts to a probability of 0.0026, for a factor of 10 it amounts to 4.7e-6. We can thus reject the box size dependence hypothesis with a relatively large statistical margin.”

2) The discussion on the distinction between thermal and kinetic stability in the first paragraph of the results section is not needed and should be removed. (The corresponding section in the reply by El Hage et al. should also be removed.) El Hage highlighted that the observed discrepancy between the experimental timescale of the T-R transition (seconds) and the much shorter simulation timescale of the transition. Although an underlying kinetic effect is also possible, it is not unreasonable to suggest that the thermodynamics of the simulation system is distorted. There is no indication that El Hage et al. are not aware of the distinction between these two scenarios. At this juncture it is a secondary issue to be concerned about whether the box size effect is of thermodynamic or kinetic nature, when the effect itself is under debate.

REPLY: We appreciate the consideration of whether this distinction is relevant when the effect itself is under debate. However, we have two reasons why we think it should nevertheless be left as a raised concern. First, leaving the issue uncommented would leave the impression that it is a valid procedure to use a kinetic readout (of non-equilibrium, unidirectional transitions) to probe the thermodynamics of a system. We argue that this is wrong and thus possibly harmful to the community, since, as we pointed out in our comment, this rests on several severe, and in this case unchecked, assumptions. As in the El Hage et al. paper a specific thermodynamic interpretation is included, we feel it is important to point this out to the readers. Not taking the opportunity to correct the scientific record on this issue would appear strangely unscientific.

Second, El Hage et al. insist that the readout is thermodynamic whereas we argue that transition times (or lack thereof) are per definition kinetic. This is true for both their and our sets of simulations and analyses. Therefore, both sets of simulations (theirs based on N=1, ours based on N=10-20) should be tested with the same tests to check for statistical significance. Without explicitly pointing out that both studies are in fact addressing kinetics (something that was also confirmed in an email from Dr. Meuwly and Dr. Karplus to us) one could easily mistakenly think that one is addressing thermodynamics and the other kinetics, and therefore different approaches are called for in both cases.

To follow the reviewers’ suggestion of the lower priority of this concern, we now made this the third rather than the first concern.

To directly address the thermodynamics (the relative stability of the T vs the R state) we have now added an analysis of transitions back to T after a T to R transition. As can be seen in supplement 1 to Fig. 1, such partial events do take place (highlighted by grey bars) and occur independently of the box size. Thus, again, the single such partial event included in the El Hage et al. reply reported for a 150A box does not significantly deviate from the distribution observed for the other box sizes, and thus does not support the assertion of a ‘stabilization’ effect of the 150A box on the T state of hemoglobin.

Changes made to the manuscript:

We have moved the thermodynamics vs kinetics concern after the other two concerns. In addition, we have added supplement 1 to Fig.1 which addresses possible transitions back to T after a transition to R had been made.

3) It is known, though rarely formally acknowledged, that when the simulation length is far shorter than the actual timescale of a molecular event, the simulation results are sensitive to the exact simulation setup. We note that the setups by El Hage et al and by Gapsys and de Groot are not identical, and the differences in the setups potentially involve several parameters (e.g., how the initial water box was constructed, how long the water molecules were equilibrated before the production runs, whether the protein system was restrained during the water equilibriation, and what types of baro-stat and thermo-stat were used et al.) To clarify these issues, please be explicit about the differences between your setups and the setups used by El Hage et al.

REPLY: We are aware of this issue and therefore had contacted El Hage et al for their setup. The data in our comment that refer to 'El Hage setup' is based on their 90A setup and therefore is identical to that of El Hage et al in all the mentioned aspects of how the initial water box was constructed, how long the water molecules were equilibrated before the production runs, how the protein system was restrained during the water equilibration, what force field was applied, and what types of barostat and thermostat et al were used. Therefore, the reviewers’ assertion above “We note that the setups by El Hage et al and by Gapsys and de Groot are not identical” appears incorrect. Based on this setup of the 90A box, we ran 20 simulations of 1 microsecond each. 9 of these did not complete the transition, and thus remain in the T state, which in the El Hage et al. 2018 study had been claimed as an exclusive feature of the 150A box. The fact that we observe the lack of a transition also in equally long simulations in the 90A box thus renders the conclusion of El Hage et al. invalid that the lack of a transition to R (or indeed a valid description of deoxyhemoglobin to remain ‘stable’ in T) is due to the larger box size in the 150A box.

In addition, we have also carried out simulations in larger boxes of cubic sizes of 120A and 150A. The 120A and 150A boxes were constructed from the 90A box by adding a water layer (with solvated ions) of 30 and 60A, respectively. These therefore also are identical in terms of protein configuration as well as protein solvation shell to the El Hage et al setup. We have now added this description to the manuscript. That the added solvation has created a setup indistinguishable from the El Hage et al setup for the 120A and 150A boxes is illustrated by the fact that the single trajectory findings of El Hage et al. fall well within the error bars that we provide not only for the 90A box but also for the 120A and 150A boxes. This can be even more clearly seen in our Fig. 1B, that shows that the El Hage et al. simulations fall well within the distributions we find for all the investigated box sizes.

In the response of El Hage et al. that was sent to us, their setup in terms of treatment of periodic boundary conditions is misrepresented:

- "Removing the angular center of mass motion is only warned against if the motion of the center of mass itself is not controlled, i.e. if the solute can cross the boundaries of the periodic box." Removing the angular center of mass (COM) motion in a periodic system always issues a warning in GROMACS. In addition, having atoms not in any COM group triggers another warning. Indeed, running with the El Hage et al. setup, two GROMACS warnings are triggered that need to be manually overridden to be able to continue. Overriding these warnings and nevertheless using this setup leads to severe issues with energy conservation. Note, that for all the hemoglobin simulations where the El Hage et al setup was used, despite the GROMACS warnings, we kept the COM motion removal exactly as described by El Hage et al, 2018.

- "the solute can cross the boundaries of the periodic box. This, however, is not the case in our simulations, but appeared to happen in some of theirs." In our simulations we follow best practices of MD with GROMACS and do not apply angular COM motion removal. Therefore, in this setup, the solute moving across the periodic box boundaries is not an issue for simulations: molecules, independent of being solvent or solute, are constantly moving across the periodic box boundaries without creating issues.

That other parameters in the setup indeed have an influence on the transition statistics is reflected in Fig. 3A where we show that three different hemoglobin setups in terms of e.g. chosen protonation states or protein-heme interactions show three different types of transition kinetics. In none of these three, we see a systematic effect of the box size, however.

Changes made to the manuscript:

We have now explicitly stated that the setup in our manuscript labeled ‘El Hage et al.’ is identical to that used by El Hage et al.: “The El Hage setup is obtained from El Hage et al and thus identical to the one used in (El Hage et al, 2018)” and “The molecular dynamics simulations for the El Hage setup were performed in cubic boxes and used the parameter set and initial structure obtained from El Hage et al. for the 9 nm box. The setup is therefore identical to that reported in (El Hage et al., 2018). The 12 and 15 nm setups were generated by adding an additional shell of water and ions around the 9 nm box. These generated 12 and 15 nm boxes behave indistinguishably from those reported by El Hage et al. (Fig. 1).” We have also pointed out that one critical difference to the other hemoglobin setups is the lack of a covalent bond between the iron and the proximal histidine.

Also, please revise your manuscript to address the following comments in the response from El Hage et al.a) The comments in the sentence that starts "A possible problem with the comparisons in their figure 3…"

REPLY: The setup issue is already addressed above. It was the choice of El Hage et al to only send us their 90A setup (we had before that shared all our setups with them). The only thing we did to construct the 120A and 150A boxes was to add a layer of solvent with ions around the pre-solvated 90A box. In the same paragraph, El Hage et al write: "In that regard, it is interesting to note that their observed life time for the 90A box with its error bars essentially includes the one observed in our simulation." The same is in fact true for the 120A and 150A boxes: the single trajectory findings of El Hage et al. fall well within the error bars that we provide. This can be even more clearly seen in our Fig. 1B, that shows that the El Hage et al. simulations fall well within the distributions we find. There is thus no reason to assume 'A possible problem' with the 120A and 150A boxes.

Changes made to the manuscript:

Please see the reply to point 3 above.

b) The comments in the sentence that starts "We also note a difference in the magnitude of the error bars…"

REPLY: the errors are different because the transition times are different. Smaller absolute transition times result in smaller associated absolute errors. We now included this notion in the revised manuscript.

Changes made to the manuscript:

“Note that the difference in size of the error bars is explained by the difference in transition times: larger transition times are accompanied by larger absolute uncertainties.”

c) The comments in the paragraph that starts "When we were still collaborating…"

REPLY: This section is about the effect of protonation states. For the systems labeled 'El Hage' in our comment, the protonation states are identical to the ones of El Hage et al (2018) as the simulations started from the input provided by El Hage et al. as pointed out above. In addition, as pointed out in our comment, the El Hage et al protonation states are nearly identical to the ones in our "Hub" setup (except for one small difference of a surface histidine of which we argue that it is unlikely to substantially affect the transition kinetics) and therefore it seems odd to state (like El Hage et al do): "protonation states of the histidine residues were very different". They are different to the Kovalevsky et al. setup, the effect of which on hemoglobin we had already reported in 2013 (Vesper and de Groot, 2013). This section in the reply of El Hage et al thus seems rather confusing and does not appear to provide novel insight. In addition, the protonation states listed in Table 1 of the El Hage et al reply differ from the setup that was sent to us.

In total, we tested 3 different hemoglobin setups with different protonation states and observed no box size dependence in any of them, as stated in the manuscript: “no dependence on the simulation box size was observed for the transition times for any of the tested setups (Figure 3).”

Changes made to the manuscript:

None, as this notion was already included in the original manuscript.

4) The comment showed that using a different analysis of the water behavior leads to a different conclusion. In the reply, El Hage et al. clarified that their attribution of the box size effect to the hydrophobic effect remains a hypothesis and it requires further investigation. Indeed, this hypothesis needs further examination by careful analyses of water structures and by extensive free energy calculations. The reviewers suggest that in the revision the authors make it clear that the hydrophobic effect hypothesis is not precluded, although the original analysis by El Hage can be improved.

REPLY: our analyses reveal that the suggested box size effect on the hydrophobic effect (as quantified by hydrogen bonding, solvent RDF or water diffusion constant), disappears when normalized taking into account the different protein-to-water ratio in the different box sizes. This 'dilution' effect is an inherent (and thus unavoidable) effect associated with an increased simulation box. As this effect is sufficient to explain the observed data, this implies there is no evidence to justify assuming an additional effect on protein solvation or the hydrophobic effect. In fact, we observe this ‘dilution’ effect also in smaller systems of which El Hage et al. argue that they would not expect a box size dependence via the hydrophobic effect (added supplementary figure 1 to Fig. 2). This is an additional indication that the analysis of RDF, hydrogen bonding and water diffusion as presented is not suitable to investigate a box size dependence of the hydrophobic effect, as the same trends are observed both in systems where the hydrophobic effect is expected to play a role and where it is not. Taken together, this thus removes the justification for Figures 3, 4 and 5 of El Hage et al (2018). Please note that this does not preclude the hydrophobic effect hypothesis, the current data however do not indicate that protein solvation or the hydrophobic effect changes with the simulation box size. We have included this notion in the revised manuscript.

Finally, there are several incorrect assertions in the response from El Hage et al. about our re-analysis of the hydrophobic effect. Some examples:

- subvolume analysis: "needs to introduce “imaginary boundary conditions” and it is unclear how this was done." No imaginary boundary conditions were involved.

- "Such an approach also appears to assume that water outside this artificial volume behaves as bulk water." This is not assumed in our approach, but explicitly tested and found to hold true.

Changes made to the manuscript:

We have added the paragraph: “As the inherent, unavoidable dilution effect associated with increasing the box size is sufficient to explain the observed data,this implies there is no evidence to justify assuming an additional effect on protein solvation or the hydrophobic effect. This thus removes the justification for Figures 3, 4 and 5 of El Hage et al. (2018). It is important to note that this does not preclude the hydrophobic effect hypothesis: however, the current data do not indicate that protein solvation or the hydrophobic effect changes significantly with the simulation box size for the studied range.”

In addition, we have added an analysis of ubiquitin (supplement 1 to Fig 2) that shows the same trend of dilution effect, which, also in this case, is not related with a change of the hydrophobic effect with the box size: “The dilution effect is evident in other systems as well: (supplement 1 to Fig. 2) demonstrates the RDF and diffusion coefficient dependence on the protein-to-water ratio in ubiquitin simulations. Note that ubiquitin is significantly smaller than hemoglobin and therefore the hydrophobic effect should be expected to be significantly smaller, following the argumentation of El Hage et al. (El Hage et al., 2018). The fact that nevertheless we see a similar box size effect on the (unnormalized) RDF and diffusion constant confirms that the underlying cause is the dilution effect rather than a box size effect on the hydrophobic effect.”

5) The box size effect, if true, is necessarily a subtle one. The authors' simulations of small proteins without observing a box size effect do not serve as proof to preclude the box size effect, especially for large systems. The authors should make this important caveat clear in their discussions.

REPLY: The claimed box size effect in the study of El Hage et al was reported solely based on sparse hemoglobin data. Based on single trajectory results from this single protein, the authors suggested implications for "existing and future simulations of a wide range of systems.", thus without limiting their findings to the single system studied or in fact a system of a specific size, but rather suggesting implications for a wide range of systems. In our comment, we have studied hemoglobin in three different variations as well as three additional systems using multiple replicas in each case. We thus cover 6 different systems as a first step to address the suggested "wide range of systems". In our comment, we have explicitly limited the validity of our findings to the systems studied by using the phrase "... for the investigated systems" in the final paragraph.

Changes made to the manuscript:

None, as in contrast to El Hage et al., we had already explicitly limited the scope of our study to the investigated systems.